



# Wave simulation for the design of an innovative quay-wall: the case of Vlora's harbour

Alessandro Antonini[1,2], Renata Archetti[2], Alberto Lamberti[2]

[1]CIRI Edilizia e Costruzioni, University of Bologna, Bologna, 40100, Italy
[2]DICAM, University of Bologna, Bologna, 40136, Italy

*Correspondence to*: R. Archetti (renata.archetti@unibo.it)

**Abstract.** The sea state and environmental sea conditions are basic data for the design of marine structures. Hindcasted wave data have been here applied for the definition of proper environmental conditions at sea, with the aim to estimate the design

condition of an innovative quay wall concept, for a more efficient dock design.

In this paper the results of a Computational Fluid Dynamic model is used to optimize the design of a non-reflective quay wall of Vlora's harbour and define design loads under the action of extreme conditions. The design wave conditions at the harbour entrance have been estimated analysing 31 years hindcasted waves data simulated through the application of WaveWatch III. Due to the particular geography and topography of the Vlora's Bay wave conditions generated from

northwest are transferred to the harbour entrance with application of a 2D spectral wave model. Southern wave states, which are also the most critical for the port structures, are defined by means of a wave generation model, according the available wind measurements. In general hindcasted sea data, after the application of wave models at a proper scale, are the basic data in support of the maritime works through the propagation of design wave condition from off-shore to harbour entrance. The results show that the proposed method based on the numerical modelling allows the identification of the best site specific

solution also for a location devoid of any wave measurement.

## 1 Introduction

The development of global trade and ship transportation often requires that the existing docks must be upgraded, consolidated or enlarged, in order to face effectively the increasing demand of people and freight traffic. With such aims, quays over piles with absorbing rubble mound slopes can be used to enlarge or rebuild structures in the existing docks.

Generally the rubble mound assures low reflection in the port basins, very important for mooring and manoeuvring but they lead to the construction of very wide superstructures that are not always possible due to the available spaces and economic sources. The use of vertical walls as berthing structures is an alternative quite used in port areas: in fact this kind of solution represents a compromise between the simplicity of construction and the small covered area. Nevertheless vertical quay walls present the drawback of undesirable high wave reflection into the port areas. Low reflecting quays sort out the reflection in

port areas by means of porous or open structures that dissipate a part of the incident wave energy. Thereby several different vertical dissipative solutions have been proposed during the last decades as result of research studies from all over the world.




The availability of wave data is a necessary condition to pursue the design of innovative harbour parts, as the quays and docks. At several locations direct measurements are not always available and series of hindcasted waves are necessary for defining the proper condition at harbour entrance. The most common operative wave generation models are WaveWatch III, WAM (WAMDI, 1988, Monbaliu et al., 2000), SWAN (Booij, 1999). WaveWatchIII model

(http://polar.ncep.noaa.gov/waves/wavewatch/wavewatch.shtml), developed by the National Weather Service (NWS) and National Oceanic and Atmospheric Administration (NOAA) is operational at DICCA, University of Genoa. The model covers the whole Mediterranean basin with a resolution equal to 0.1° (Sartini et al., 2014, Sartini et al. 2015). The model chain is forced with the National Centers for Environmental Prediction (NCEP) Climate Forecast System Reanalysis (CFSR) data at 0.5° resolution. Validation of the model has been extensively performed (Mentaschi et al., 2015). On the basis of such

numerical results, the design wave conditions are estimated through extreme wave analysis. Extreme events analysis can be based on a widely number of methodology (Mazas et al., 2014, Masina et al, 2015) while the historic coastal structures design practice leads to the use of Peak Over Threshold technique to select samples data (Mathiesen et al., 1994, Garcia-Espinel et al., 2015).

To attenuate wave reflection various structures have been designed, being Jarlan-type structures (Jarlan, 1961) the most

15 widely used. However, all the existing antireflective solutions for vertical maritime structures have the drawback of their exiguous efficacy to reduce the reflection of low frequency waves (i.e. wave periods larger than 25 s). To overcome this technical problem, the design of a vertical structure can be based on a multi-cell circuit concept which is considered to be especially effective to reduce the wave reflection of wind waves and oscillations associated with intense storms, resonance waves in port basins, (Medina et al., 2010, Garrido et al 2014). Altomare and Gironella (2014), Matteotti (1991) and Faraci

et al. (2012) investigated, by means of physical model tests, quay walls consisting in prefabricated caissons with frontal openings and internal rubble mounds. In cases where the sheet piling is allowed, when great retaining heights have to be achieved a combined quay wall structure is normally preferred for its structural lightness.

In this work we present an application on the use of hindcasted data for defining the proper design condition at the entrance of a new harbour in the Vlora's bay; the results of extreme analysis is fundamental input for the optimisation of a new

concept of quay wall. The quay wall is investigated, by means of CFD simulations CD-ADAPCO (2013), Lamberti et al., (2015). The structure is consisting of a sheet-pile quay-wall, a relieving platform, an anchors and piles foundation system and an antireflective wave chamber filled with a rock-armoured slope.

Aim of the paper is twofold, *i)* showing application of sea situation estimation, through the use of wave hindcasted data, in order to properly define the design wave conditions for a new harbour, *ii)* presenting new approach for investigating

innovative non-reflective quay-wall.

## 2 Site description

The city of Vlora, Republic of Albania, is located on a coastal plain in the North East part of the Bay. The Bay is approximately 17 km long and 10 km wide with a depth that reaches 55 m. It is open to the Adriatic through North-West



side, while is bordered by tall and craggy mountain peaks to the south and west. The bay is also partially protected from the swells coming from north-west by the presence of Sazan Island, located at the mouth of the bay, and from the shallow water next to the mouth of the river Voiussa. The investigated port is located in the small northern stretch of the coast in the inner part of the Bay. The location of the new harbour is presented on Fig. 1 by the yellow star.

In the surrounding area there are already some port infrastructures, some of them are still working and other are disused. Such facilities include an offshore berth for tankers called New Port and the historical Old Port of Vlora. The harbour under investigation presents two piers as shown in Fig.2, west wharf and east pier. The first is primarily intended for docking ferry passengers while the second is used for berthing cargo vessels.

**2 Design wave conditions**

No measured wave data are available for Vlora's bay, therefore wave climate was defined by means of 31 years (1999 - 2010) of hindcasted wave data supplied by the University of Genova, (Sartini, 2015, Mentaschi et al., 2015). The extraction point (red star in Fig. 1) was selected on a grid point, representative of the wave climate outside the gulf. The wave rose at the extraction point is presented in Fig. 3.

Two main directional sectors are evident: 240°N - 40°N, associated with Adriatic axis direction, and 160°N –200°N,
providing information on southern waves. Dominant wave conditions, characterized by the highest wave heights, are related to the sector between 155°N -170°N whereas the maximum frequency of northern events, is related to 300 ° N and is equal to 15%. Maximum northerly wave height is equal to 5.75 m, with 10 s peak period.

Due to the particular geography and topography of the Bay, southern wave data are not representative of the actual conditions in the vicinity of the port. Consequently, definition of southern wave states (S-SW) requires another source of
information like wind measurements that are partially available inside the Bay.

In order to estimate the design wave conditions reaching the harbor entrance, two different approaches have been undertaken for waves propagating from the Northern sectors and for waves generated by the S-SW wind. For both main directions, two years ($T_R$=2) and one hundred years ($T_R$=100) return period wave states were estimated. Two years return period, event is considered the limit of ordinary operation, that is, the limit condition for which there should not be any malfunction of the
harbor infrastructure, while 100 years return period, event is the design condition of the structures, namely, the event for which the structures should only resist.

The northern wave conditions are established by the extreme wave analysis, performed on the hindcast wave data. The southern design wave conditions have been estimated by applying the wave generation module of Mike21-SW on the Vlora's bay, by imposing as a main input the local wind intensity with $T_R$= 2 years and $T_R$= 100 years.

**2.1 Design Waves from NW**

The peak over threshold (POT) method has been applied to the northern storms time series (N-NW) in order to select independent storm peaks. The technique is a well-established statistic approach, based on the following steps: *i)* application





of a proper threshold; *ii)* selection of homogeneous independent events and peaks which exceeding the selected threshold; *iii)* identification of the probability model that best represents the exceedances; *iv)* determination of values within a given return period. The great advantage of POT method is the utilization of an adequate number of independent extreme data, achievable also with a relatively short time series.

The POT analysis has been applied to the wave data providing the results illustrated in Fig. 4, where the empirical frequency histogram and cumulative distribution function (pdf and CDF) are given along with the corresponding Gumbel distribution, with best fit scale parameter ($\sigma$) and location parameter ($\mu$) equal to 0.58 (95% lower and upper bounds parameter fitting: $0.55 - 0.61$), 2.51 (95% lower and upper bounds parameter fitting: $2.48 - 2.55$), respectively. The threshold used to define individual storm has been established according to Boccotti's method, (Boccotti, 2000). The method is based on a

preliminary identification of a wave height threshold, namely 1.5 times the annual average $H_{Sy}$, that is 0.5 m for the available wave data: the chosen threshold value is therefore equal to 1.50 m. The application of the described approach led to the detection of 355 independent events from the direction -120°N +40°N.

The analyses have provided the significant wave height (Hs) and the mean peak period (Tp), whose values are given in Table 1 with reference to a return period of 2 years and 100 years. The mean peak periods associated with the predicted extreme

significant wave heights have been estimated using a scatter plot of measured peak period versus significant wave height as proposed by Viselli, (2015) and Schweizer et al. (2016). The parameters a and b in eq. 1. turn out to be equal to 2.36 (0.30, 4.42) and 0.72 (0.43, 1.01), respectively, values in parentheses indicate the lower and upper bounds of the 95% confidence interval.

$$T_p = a \cdot H_s^b \qquad (1)$$

Not exceedance probability distribution which best fits the extreme events data is the Gumbel distribution, the resulting values of $H_S$ for the selected wave height are listed in Tab. 1. It is worst to remark that the estimated thirty years return periods ($T_R$=30 years) significant wave height ($H_{TR30}$) is equal to 5.8 m, in accordance with the maximum hindcast wave height equal to 5.75 m.

**2.1.1 Propagation of design waves at the harbor entrance**

Numerical model Mike 21-SW was applied to propagate the conditions from the offshore location to the entrance of the port. The spectral wave module simulates the growth, decay and transformation of wind-generated waves. MIKE21-SW is third-generation spectral wave models, as it doesn't require any parameterization on either the spectral or the directional distribution of power (or action density). The physical processes modelled comprise: (*a*) energy source/dissipation processes

(wind driven interactions with atmosphere, dissipation through wave breaking / whitecapping / wave-blocking due to strong opposing currents, bottom friction-induced dissipation), (*b*) non-linear energy transfer conservative processes (resonant quadruplet interactions, triad interactions), and (*c*) wave 20 propagation-related processes (wave propagation due to the wave group / current velocity, depth-/current- induced refraction, shoaling, interactions with unsteady currents). The models



compute the evolution of wave action density by solving the action balance equation as described by Booij et al., (1999). MIKE21 modelling suites discretize the computational domain by unstructured triangular meshes (flexible mesh).

Recent applications of MIKE 21-SW with a flexible mesh are described in Samaras et al, (2016) and Gaeta et al., (2016). The bathymetric and shoreline data used in this work resulted from the digitization of nautical charts acquired from the

5    Italian National Hydrographic Military Service ("Istituto Idrografico della Marina Militare"). The triangular mesh dimension is homogenous over the entire domain, resulting in a mesh forming 102995 elements with dimension of approx. 600 m$^2$. The grid for the entire domain, and harbor entrance is shown in Fig.5.

Wave field of $H_{TR2}$ and $H_{TR100}$ are presented in Fig. 6 and Fig. 7. In which reduction of the wave height due to the shoaling is shown. The design conditions at the harbor entrance, on a -13.5 m depth (UTM coord. 371221 E, 4478470 N, Fig. 4) are

10    presented in


Table 2.

### 2.2. Design Waves from S

Southern design wave conditions were not easily estimated, as direct measures of waves inside the Vlora's bay are not available and also wind data are relatively scarce. Such conditions, which are also the most critical for the port structures, are
defined by means wave generation model of MIKE21-SW.

The approach is well known and here shortly summarised: Wind-wave generation is the process by which the wind transfers energy into the water body for generating waves. The wind input is based on Janssen's (1989, 1991) quasi linear theory of wind–wave generation, where the momentum transfer from the wind to the sea not only depends on the wind stress, but also to the sea state itself. The non-linear energy transfer is approximated by the DIA approach Hasselmann et al.
(1985). The source function describing the dissipation due to white capping is based on the theory of Hasselmann et al. (1985) and Janssen (1989). The source function describing the bottom induced wave breaking is based on the well proven approach of Battjes and Janssen (1978) and Eldeberky and Battjes (1996). A detail description of the various source functions is available in Komen et al. (1994) and Sorensen et al. (2003). The default source of wind for the modelling of waves is similar to the source functions implemented in the WAM Cycle 4 model, Komen et al. (1994). More details on the
wave generation module are provided in the DHI Manual (2011). The wave generation model was forced by the wind conditions with $T_{R2}$ and $T_{R100}$ (see Tab. 3, Lamberti et al., (2015)) with Charnock parameter equal to 0.1 and default input in the model.

Wind data measurements were supplied by SIAP-MICROS s.r.l., the anemometer was installed on the Albania coast at point of coordinate 40°30'51,98" N, 19°23'36,69" E, presented in Fig. 1 with a blue star. The measurement protocol followed the
standard, measuring every 10 minutes, the instrument measured for 2 years (July 2006-August 2008). The statistical analysis of the data was preceded by a quality control check of all data to remove the outliers and to interpolate over small data gaps that may be present. Overall, the corrected data were of sufficient quality, with less than 3% of the data removed as outliers or unacceptable data, Belu and Koraicn (2009). Highest measured value is equal to 22.8 m/s, while a gust reduction factor equal to 0.96 is applied in order to define the effective wind velocity, El-Hawary (2000). Fig.8 shows the wind rose based on
2 years data.

The wave fields generated by the wind conditions in Tab. 3 are presented in Fig. 9 and Fig. 10, while the extracted waves conditions at the harbor entrance are presented in Tab. 4. Final result of the design wave characterization is a set of four wave states, accounting two different directions (i.e. north-west and south-west) and two return periods (i.e. 2 and 100 years), Table 4.

### 3. Optimisation of the quay wall design

Quay-wall structure, consisting of steel piles foundation system and antireflective rock armoured wave chamber is investigated through CFD model by means of commercial code STAR-CCM+ whose has been widely used for practical





design approach, i.e. see Antonini et al. (2016a) validated with data in Antonini et al., (2015). An example of this innovative design is presented in Fig. 11. Optimum site-specific geometry is defined through the comparison of three geometries. The changes made during the analysis are related to:

1.  length of the absorbing cell;
2.  extension of the gap between front reflective surface and free surface;
3.  arrangement of the armour slope inside the cell.

Aim of site-specific absorbing quay-wall design phase is the maximization of incident wave energy dissipation according the exercise wave condition (described in Tab.4). The goal is reached considering a cell geometry with reflective surfaces far as close as possible to the quarter of the wavelength, therefore only a single wave state (i.e. southern exercise condition) is selected to optimize the geometry of the cell. In this light, for the analysis that follow the optimization of the absorbing cell was carried out based on the characteristics of the wave climate recognized as ordinary storm from the South-West, i.e. wave n° 3, ($H_s$ = 1.80 m, $H_{max}$ = 3.2 m, $T_p$ = 4.5 sec, $T_s$ = 4.2 sec), while for waves n° 1 and 4 the performances of the optimized cell are investigated.

Geometry of structure 1 has a cell length equal to 5.47 m and the front reflective surface extends from -7.5 m to -2.5 m. Armor slope extends from the front sheet pile up to the rear bean installed on top of back sheet pile, with a the nominal diameter ($D_{N50}$) equal to 1.00 m (Fig. 12).

Structure 2 has cell length equal to 5.80 m, and the front reflective surface extends from -7.5 to -3.0 m. Armor slope extends horizontally from the front sheet pile up to 2.70 m inside the cell where a slope of 1:4 reach the back sheet pile, nominal diameter is 1.00 m. An under layer is designed in order to separate the rocks and the bottom sand. Its nominal diameter is 0.50 m (Fig. 13).

Structure 3 presents the same geometrical characteristics of structure 2 in terms of length and depth of the cell but some differences in the arrangement of the armor slope which extends from the front sheet pile up to the back sheet pile with a slope equal to 1:2, nominal diameter is 1.00 m (Fig. 14).

**3.1 CFD numerical model set-up**

A k–ω SST turbulence model (Menter, 1994) is applied with a two-layer all y+ wall treatment model, and a second order implicit scheme was utilized for time marching. The transient SIMPLE algorithm is applied to linearize the equations and to achieve pressure–velocity coupling. A volume of fluid method (VOF) is applied to describe the free surface. The calculation is performed on a fixed grid and free surface interface orientation and shape are calculated as a function of the volume fraction of the respective fluid within a control volume, Antonini et al., (2016b).

A right-handed Cartesian coordinate system is located at the intersection between frontal structure surface, the undisturbed water surface and the medium vertical section of the domain. The longitudinal x-axis is pointing towards the outlet boundary, the z-axis is vertical and points upwards, and the undisturbed free surface is the plane z=0 (Figure 15:).



The domain region is 2.62 m wide (-1.31 m ≤ y ≤ 1.31 m, i.e. structure pile center to pile center distance), 38.75 m high (-7.75 m ≤ z ≤ 31.0 m) and its length varies according to the simulated wave group length ($\lambda_g$), (i.e. - $3/4\lambda_g$ ≤ x ≤ 6). The seabed is given 7.5 m below the mean water surface, and a superposition of linear waves velocity profile is specified at the up-wave boundary in order to generate an extreme focused wave group at the structure location according the design wave

climate defined above. The pressure outlet is implemented at the down-wave domain boundary. Four boundary conditions have been used to describe the fluid field at the domain bounds. They involve: no-slip wall, velocity inlet, pressure outlet and symmetry plane condition. No-slip wall boundary condition represents an impenetrable, no-slip condition for viscous flow, such a boundary is used to describe the structure surface and the bottom (z=-7.5 m). Velocity inlet boundary represents the inlet of the domain at which the flow velocity is known according to the required wave profile, this condition is used to

model the up-wave boundary at $x$ =-3/4·$\lambda_g$ and the top (z=31.0 m) of the domain, while lateral boundaries (y=-1.31 and y=1.31 m) are discretized by means of symmetry plane. The pressure outlet boundary is a flow outlet boundary for which the pressure is specified, in this model we used condition of calm water surface, outlet boundary is imposed at the boundary over the structure ($x$ =6 m) only for the air phase.

**3.2 Modelling of the armor slope**

The armor slopes proposed, within the absorbing cell, are modelled through the insertion of two solid communicating with the fluid field. By means of these volumes, dissipation due to the inertial and viscous forces are imposed. These two terms are subject of a long-time study as demonstrated by several authors (Burcharth and Andersen, 1995, Cruz et al., 1997, Engelund, 1953) who proposed different methods to calculate them according the size and type of the modelled rocks. According the Burcharth's method and considering a nominal diameter of the armour rocks equal to 1.0 m the adopted

values in this study are 6.4 kg/m$^3$/s for viscous dissipation coefficient and 1221 kg/m$^4$ for the inertial coefficient, while for the under layer characterized by a nominal diameter equal to 0.5 m the adopted values are 26 kg/m$^3$/s for viscous dissipation coefficient and 2440 kg/m$^4$ for the inertial one, adopted porosity value is equal to 0.38 and is kept constant for both the porous domains.

**3.3 Simulated wave conditions**

Three different wave conditions have been simulated, two characteristics of operating conditions, (wave n°1 and wave n°3 in Tab. 4) and one dealing with extreme conditions (wave n°4 in Tab. 4). We assume that the modelling of extreme condition from North-West (i.e. wave n° 2) is not necessary as the wave state is significantly less intense than southern one, as highlighted in the wave climate study. In order to reduce the computation time, wave groups according to a Jonswap spectrum are imposed at the inlet boundary. The wave groups are generated by a superposition of eight linear components,

each of them characterized by its own amplitude, period and phase. The input signal is calculated according the Boccotti's theory, (Boccotti,1983 and Boccotti et al. 1993). The procedure allows defining an extreme focused wave group coherent with real sea state described by a spectral distribution. In this study a Jonswap spectral shape equal to 3.3 is adopted, while



the maximum wave height at the focusing point is identified through the Goda's method, Goda, (2000), (i.e. wave n° 1 $H_{max}$=1.80 m; wave n° 3 $H_{max}$=3.24 m; wave n° 4 $H_{max}$=5.40 m, Fig. 17).

**3.4 Grid generation**

The domain mesh and prism layer grids are generated using the mesh generator in STAR-CCM+. Grid resolution is finer near the free surface and around the quay-wall structure to capture both the wave dynamics and the details of the flow around the structure, Fig. 16. Prism-layer cells are generated along the structure's surface, the height of the first layer is set so that the value of y+ (10 to 400) satisfies the turbulence model requirement by solving the velocity distribution outside the viscous sub-layer, i.e. buffer layer and log-law regions are solved, Demirel et al. (2014), Schultz and Swain (2000). Regular hexahedral cells discretize whole domain, while four thinner areas are used to capture free surface movements and the interaction between the wave group and the structure, ($V_{w1}$, $V_{w2}$, $V_{S1}$, $V_{S2}$). The grid refinements across the water surface are realized by the volumetric controls $V_{W1}$ and $V_{W2}$ proposed along the whole domain. $V_{W2}$'s height is equal to the maximum simulated wave height while $V_{W1}$'s height is 50% more (see longitudinal section in Fig. 16). $V_{W2}$'s horizontal grid size is determined by the wavelength of the shortest linear incident component ($\lambda_{i1}$), i.e. $\Delta x = \Delta y = \lambda_{i1}/60$, while vertical grid size is adjusted according to the wave height of smallest linear incident component ($H_{i1}$), $\Delta z = H_{i1}/20$, $V_{W1}$'s grid sizes are 50 % larger than $V_{W2}$'s ones. Volumetric controls used for the refinements around the structure present the same grid dimension of those used to discretized the water surface but their dimension are set in order to include a gap between the structure surface and volumetric control edges equal to 1.0 m for $V_{S1}$, and 5.0 m for $V_{S2}$. These mesh characteristics contribute to generate a grid of variable cell numbers from $2.0 \cdot 10^6$ to $3.5 \cdot 10^6$ according to the incident wave condition. To assure the numerical stability and respect the requested Courant number, a time step equal to $T_p/400$ is adopted in the study. All the RANS simulations are carried out on the work-station at the hydraulic laboratory of the University of Bologna, double compute node consisting of hexa-core 2.00 GHz Intel Xeon E5. For a mesh with $2.0 \cdot 10^6$ elements, it takes about 36 h on 12 cores to complete the entire wave group attack.



### 4. CFD results

Results are presented in terms of mean values of reflection coefficient ($c_r$), and pressures acting on the most significant parts of the structure. Pressures are analysed only with respect to the extreme conditions. Surface elevation is measured through 6 wave gauges placed at the domain axis (y = 0) and at different distances from the quay-wall, (x = -18; -16; -14; -12; -10; - 8

5  m). The selected range of distance values are chosen in order to be at least far from the quay wall for 1/4 of the incident peak wavelength in order to exclude the effects of stationary oscillations which cannot be explained through the adopted reflection analysis method, Zelt & Skjelbreia, (1992).

### 4.1 Reflection coefficients results

Wave reflection induced by the quay-wall is quantified by the reflection coefficient, defined as follows:

$$c_r = \frac{\sqrt{m_{0i}}}{\sqrt{m_{0r}}} = \frac{H_i}{H_r};$$ (2)

$$m_0 = \int_{0.5/f_p}^{1.5/f_p} S_{(f)} \cdot f^0 \cdot df$$ (3)

where $H_r$ and $H_i$ are the reflected and incident wave heights, respectively, and $m_{0r}$ and $m_{0i}$ are the related zero-order spectral moment calculated between 0.5 and 1.5 times the peak frequency.

Calculation of the reflection coefficient is carried through spectral analysis of a non-stationary signal, since the generated

wave group is represented by a short time series coherently defined according the above identified wave states. The approach implies the use of a time fixed window (i.e. fixed position along the time series vector) centred on both incident and reflected wave group. A trapezoidal shaped window is used to reach this scope. Its length is defined according the signals shape, i.e. the window begins with the first value above 0.05 m identified within the incident wave and closes after the last value above 0.05 m identified within the reflected signal. Two linear slopes between 0 and 1 characterize the window shape. The slopes

length is equal to 5% of the groups envelope (Fig. 18). According to this approach the reflection phenomenon is assumed to be linear, thus reflected and incident spectral shape will not change except for the reduction of energy that is the investigated parameter.

Comparison of the results enables to identify the optimal geometry as the structure 3, proposed in Fig. 19. It can be noted that there is a generalized reduction of the reflected wave energy for whole range of analysed frequencies.

With regard to the lower frequencies, i.e. incident wave periods longer than 4.5 s, the required wave attenuation cannot be guaranteed only by the resonance phenomenon of the cell, in this light the presence of the rocks inside the cell become more important. The arrangement of the geometry with the smaller amour slope (structure 2, red line in Fig. 19), does not induce the expected general improvement, whereas it is important for structure 3.





With regard to the higher frequencies, i.e. incident wave periods shorter than 4.5 s, the significant improvement associated with the structure 3 is due to the variable length of the absorbing cell, generated by the inner slope that faces to the incoming waves. With regard to the central frequencies, i.e. incident wave periods around 4.5 s, the largest improvement happens between structures 1 and 2 because of the increased characteristic length of the absorbing cell. Structure 1 presents a ratio

between incident wave length and its own length equal to 0.18 while for structure 2 is 0.195, facilitating the establishing of the resonance condition inside the cell.

A synthesis of the reflection coefficients is presented in Tab.5. It can be considered satisfactory the averaged reflection coefficient determined for the structure 3. Therefore, structures 3, (presented in Fig. 11), can guarantee the required internal agitation levels and it has been selected as the optimum structure for the future quay-wall of Vlora's harbour.

**4.2 Pressures results**

This section shows the pressures acting on different parts of the structure. In order to validate pressures and forces acting on the quay-wall, numerical results are compared with the values calculated by means of empirical formulas based on physical model tests carried out by McConnell et al. (2004), see Fig. 20. a/b/c, whereas the pressures distribution on the front sheet pile is compared with the distribution calculated by means of Goda's method, (Goda, 2010), see Fig. 20. d. Four design

pressures are identified according the selected structure main parts. Eight virtual wave gauges are used to measure the uplift pressure acting on the frontal beam, Fig. 20.a; twenty virtual wave gauges are used to measure the uplift pressure acting on the internal beam, Fig. 20.b; four wave gauges are used to measure the horizontal unitary force acting on the frontal beam, Fig. 20.c and finally sixteen wave gauges are used to measure the pressure distribution on the frontal sheet-pile, Fig. 20.d. Two main activities have been carried out in order to reduce the effect of high frequency loads, and at the same time

consider all wave gauges positioned on the single investigated structure area. Firstly, the average pressure signal is calculated except frontal sheet-pile pressure distribution, which is defined through the single gauge signal. Secondly, low-band filter is applied considering a cutting frequency equal to 2 Hz, which is roughly the natural period of the structures analysed. The considered design pressure values are the maximum value of each filtered signal, while pressure distribution on the sheet-pile is calculated according the maximum value of each single pressure signal, (Tab. 6).

Each value arises from the analysis completed on all wave gauges positioned on the single investigated structure area.

The average signal obtained from the different probes has been filtered in order to exclude effect of high frequency loads. The adopted design pressure values are the maximum value of the filtered signal (Table 6).



### 5. Conclusion

Hindcasted wave data have been here applied for the definition of environmental conditions at sea, with the aim to estimate the proper design conditions for an efficient dock design based on an innovative quay-wall concept.

Design wave conditions are identified on the basis of the wind and wave data hindcasted outside and through the bay of

Vlora. Because of the different directions and periods, the identified waves conditions are distinguished in southerly and northerly waves. According to this classification two limit conditions were assessed: the limit of the ordinary conditions for which the service condition has to be ensured, and the extreme limit for which the only requirement is the resistance of the structures. Once the wave conditions have been identified the analysis of the structures performance in terms of reflection coefficients is carried out by means of CFD code. The optimization of the absorbing cell is carried out only for the ordinary

conditions, while the complete reflection analysis is completed through the other two identified wave states. The resulting structure is a quay-wall with a non-reflective cell characterized by a ratio between wavelength and cell length equal to 0.195 with a 1:2 armor slope realized on the entire absorbing cell extension. The main findings are three values of reflection coefficients $c_r$ 0.56, 0.33 and 0.4 for wave n° 1, 3 and 4 respectively. With the same code the pressures acting on the main structural parts of the quay-wall are evaluated under the southern extreme conditions. Furthermore, a comparison of the

numerical results with empirical formulas is proposed in order to validate the calculated pressure values. A general good agreement for the results is recognized through the compared pressure values. In conclusion sea data strongly supported a correct optimisation of engineering design at Vlora's harbor.

### Acknowledgements

The authors gratefully acknowledge Prof. Giovanni Besio, Carlo Zumaglini, SIAP-MICROS s.r.l., for providing respectively

waves and wind data, Piacentini S.p.A. for the discussion on the quay design and Andrea Pedroncini DHI Italia for the useful discussion on the MIKE21.

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





**Table 1: Extreme events interpretation of the northern wave data hindcasted at the selected location off the Vlora's bay**

|  | *Hs [m]* | *Tp [s]* | *Dir [°N]* |
|---|---|---|---|
| *Tr=2 years* | *4.40* | *8.5* | *310* |
| *Tr=100 years* | *6.85* | *11.0* | *310* |



**Table 2: Wave conditions extracted at the harbor entrance (UTM coord.  371221 E, 4478470 N)**

|  | *Hs [m]* | *Tp [s]* | *Dir [°N]* |
|---|---|---|---|
| *Tr=2 years* | *1.00* | *8.5* | *260* |
| *Tr=100 years* | *1.70* | *11.0* | *265* |





**Table 3: Wind conditions imposed as input to the MIKE21-SW wave generation module.**

|  | U [m/s] | Dir [°N] |
|---|---|---|
| Tr=2 years | 22 | 215 |
| Tr=100 years | 41 | 215 |





**Table 4: Selected wave states**

| Wave n° | $T_R$ (years) | Hs (m) | Tp (s) | Lenght (m) | Dir (°N) |
|---|---|---|---|---|---|
| 1 | 2 | 1.0 | 8.5 | 85.5 | 260 |
| 2 | 100 | 1.70 | 11.0 | 117.0 | 265 |
| 3 | 2 | 1.8 | 4.5 | 31.3 | 215 |
| 4 | 100 | 3.8 | 6.1 | 53.4 | 215 |





**Tab 5: Calculated reflection coefficients**

| Structure | Wave n° | Reflection Coeff. ($c_r$) |
|---|---|---|
| Structure 1 | Wave n° 3 | 0.38 |
| Structure 2 | Wave n° 3 | 0.36 |
| Structure 3 | Wave n° 3 | 0.33 |
| Structure 3 | Wave n° 1 | 0.56 |
| Structure 3 | Wave n° 4 | 0.40 |



**Tab 6:. Calculated design loads**

| Structure | Wave n° | Pressure - Force |
|---|---|---|
| Up-lift frontal pressure | Wave n° 3 | 28.5 kPa |
| Up-lift internal pressure | Wave n° 3 | 24.5 kPa |
| Frontal unitary force | Wave n° 3 | 17.0 kN/m |
| Sheet-pile | Wave n° 3 | Presure profile described in Fig. 20 |



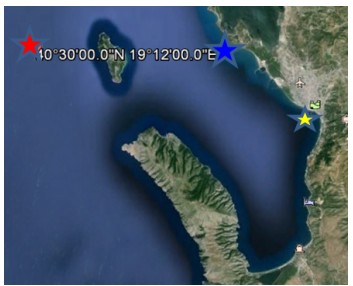

**Figure 1: Wave hindcast extraction point (Red star), Vlora's harbour (yellow star), wind data measurement point (blue star).**





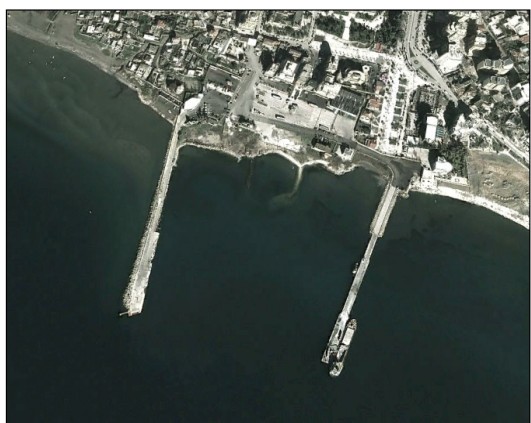

**Figure 2: Actual plan view of the Vlora's Port**





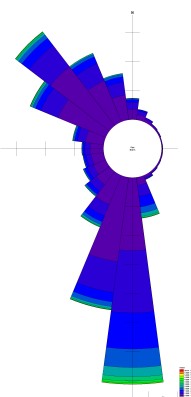

**Figure 3: Wave rose of the significant wave heights H$_S$ at the entrance of Vlora's bay (coord. 40°30' N, 19°12' E) with hindcasted data**





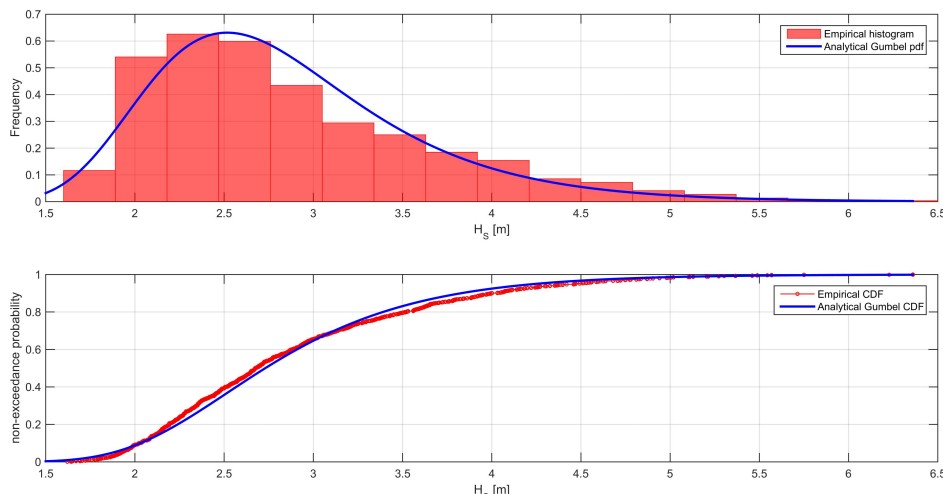

**Figure 4: Interpretation of the wave data at the study area: a) empirical and analytical pdf and b) empirical and analytical CDF
5    for the identified peak values**





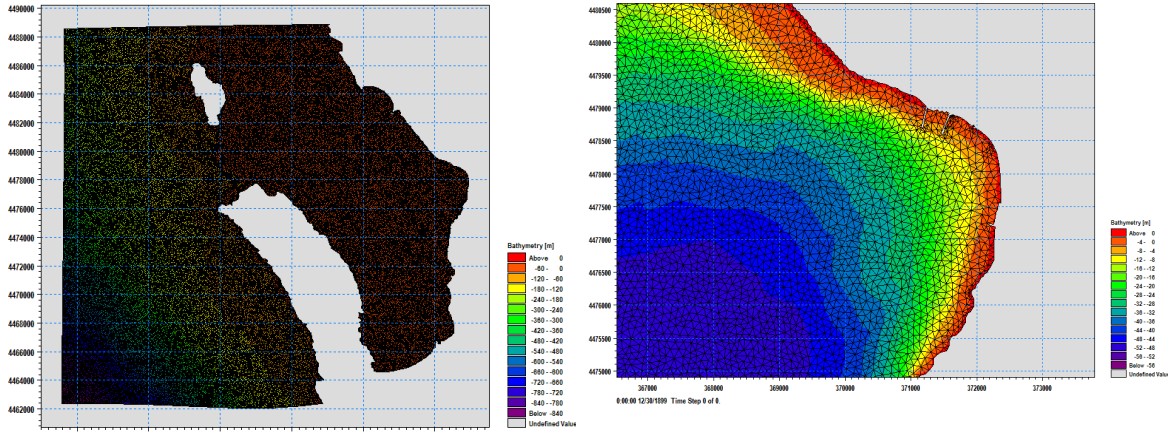

**Figure 5: Mesh of the wave model (left) and zoom to the harbor entrance area (right).**




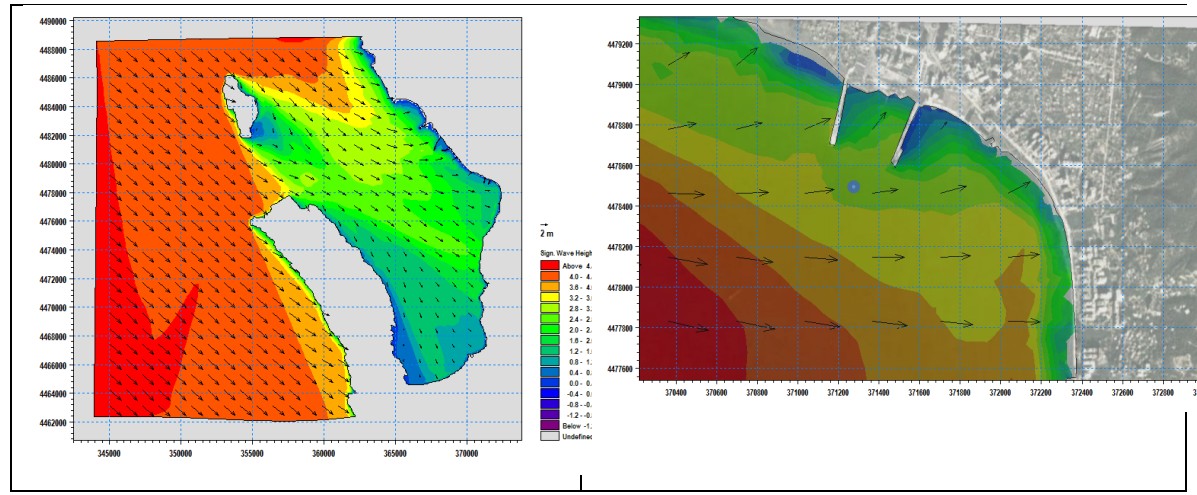

**Figure 6: Wave field for design wave condition Hs=4.40m, Tp=8.5s, Dir =310°N. The blue bullet is the extraction point location, its UTM coordinates are 371221 E, 4478470 N**





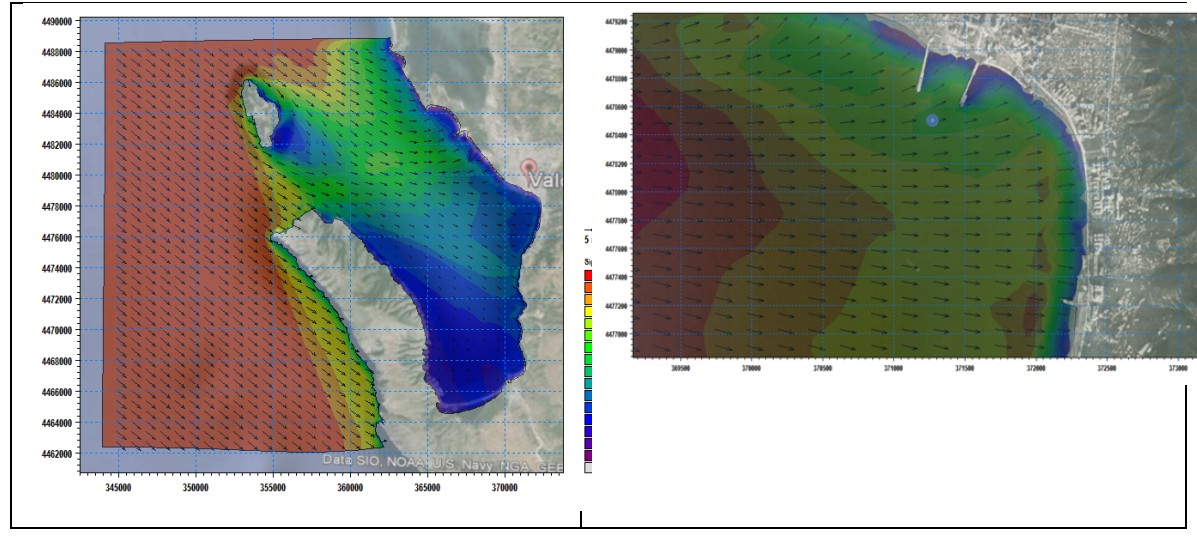

**Figure 7: Wave field for design wave condition Hs=6.85, Tp=11, Dir =310°N. The blue bullet is the extraction point location, its UTM coordinates are 371221 E, 4478470 N**





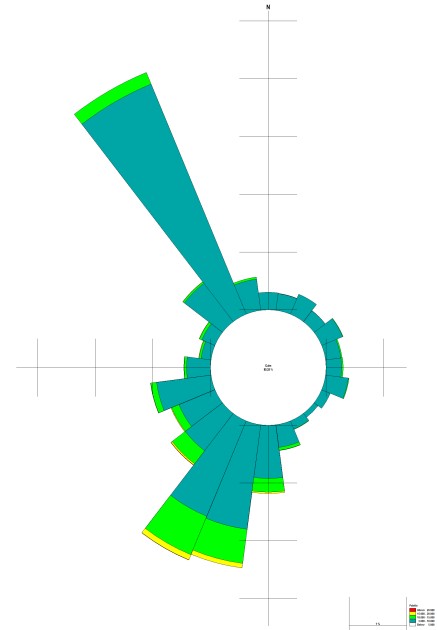

**Figure 8: Wind rose of the wind intensity. (July 2006–August 2008, SIAP s.r.l. data).**




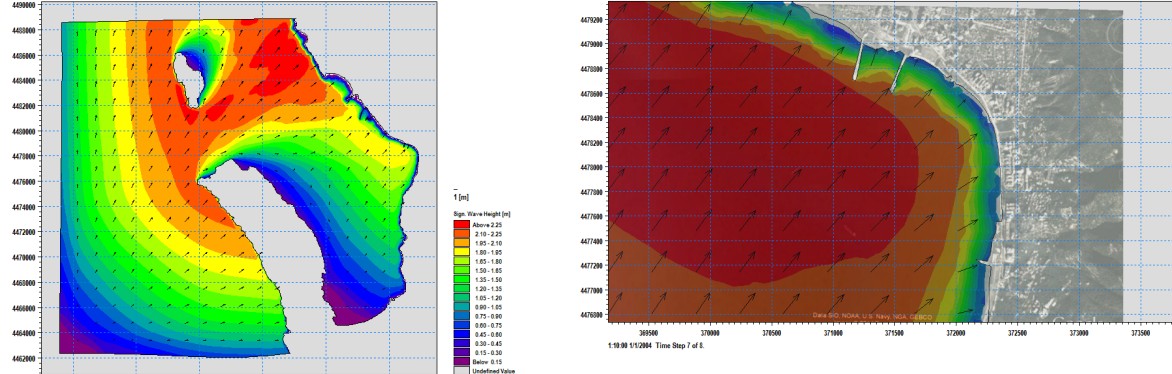

**Figure 9: Wave field for design wave condition generated by wind U=22 m/s, Dir 215°N. $T_R$=2 years**




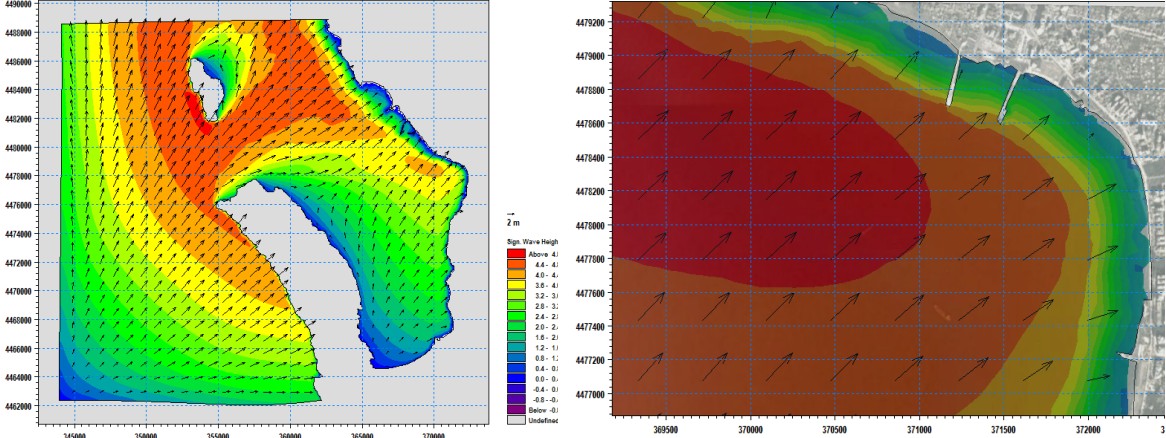

**Figure 10: Wave field for design wave condition generated by wind U=41 m/s, Dir 215°N. T$_R$=100 years**





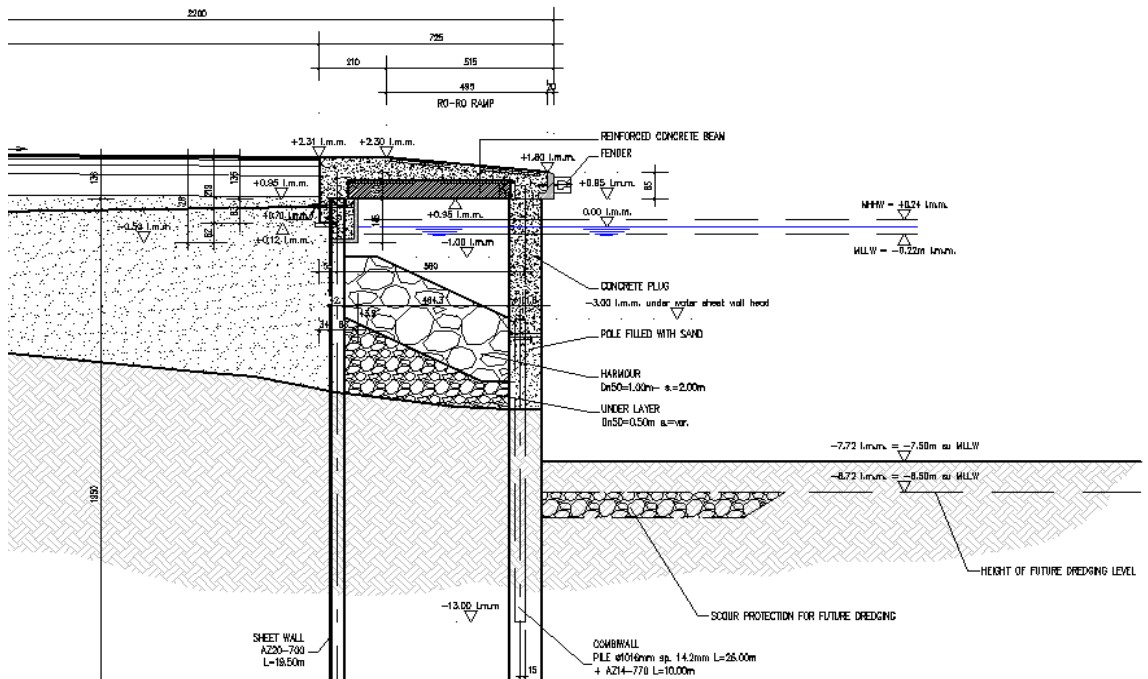

**Figure 11: Innovative quay wall section**





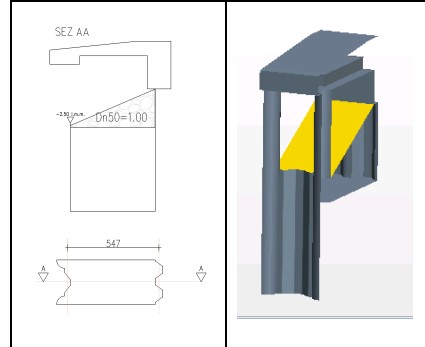

**Figure 12: Simplified section and representation for structure 1**





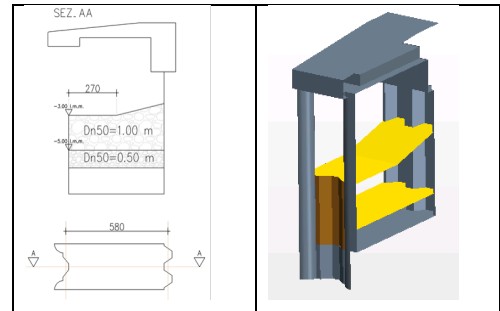

**Figure 13:   Simplified section and cad representation for structure 2**





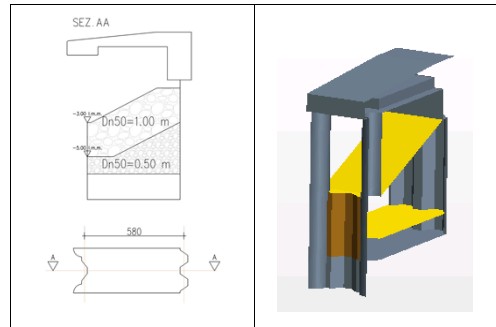

**Figure 14 Simplified section and cad representation for structure 3**

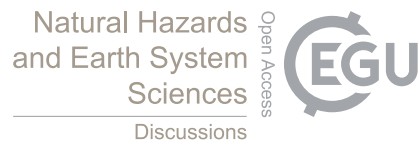

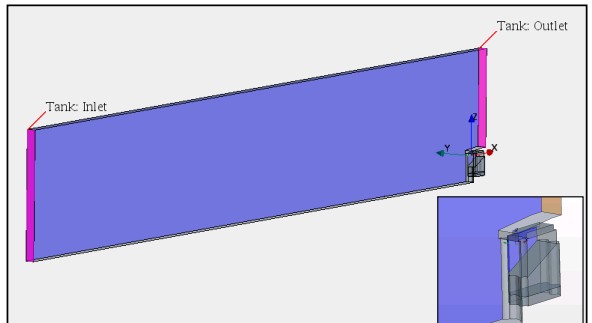

**Figure 15: Domain, boundary conditions and structure detail**





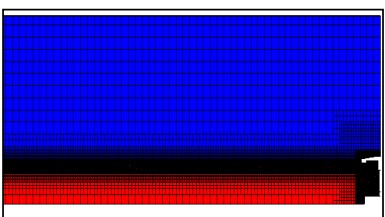

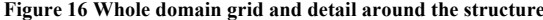

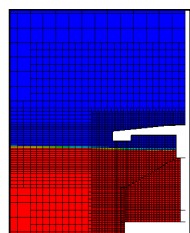

**Figure 16 Whole domain grid and detail around the structure**





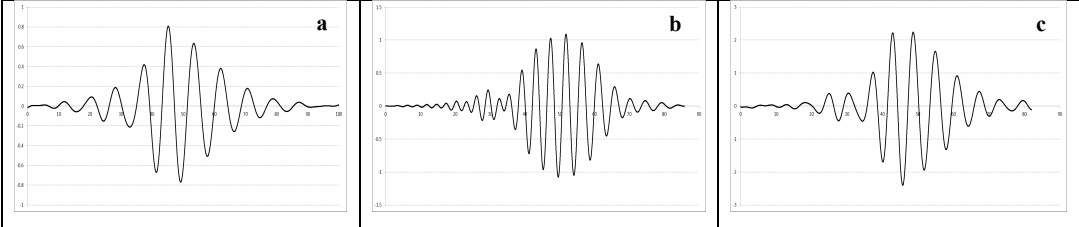

**Figure 17 Imposed surface elevation at the generation boundary: a) Wave n° 1, b) Wave n° 3,  c) Wave n° 4**



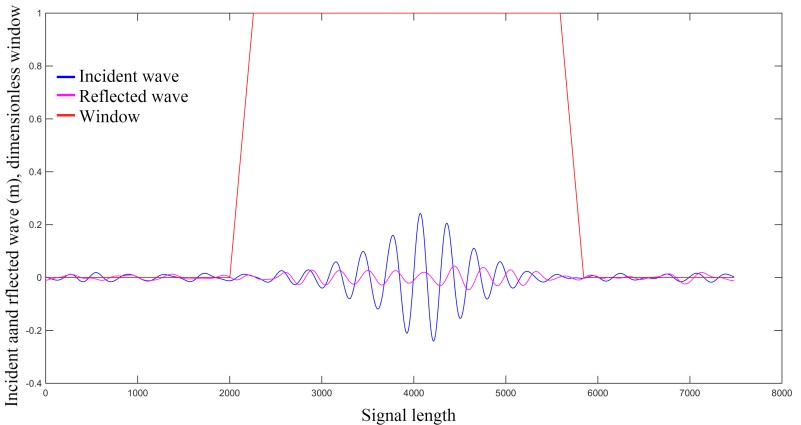

**Figure 18 Example of the adopted window**

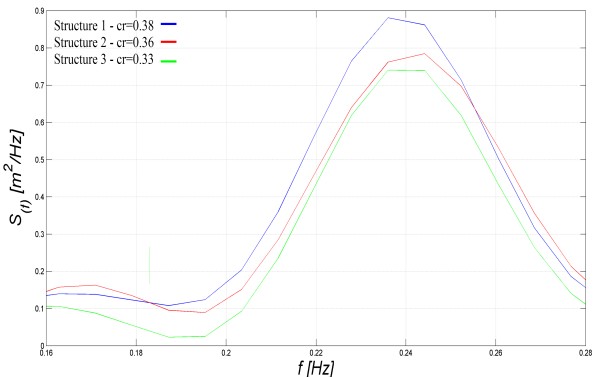

**Figure 19 Comparison of the reflected wave spectrum for wave n° 3 Tab 7:. . Calculated reflection coefficients**



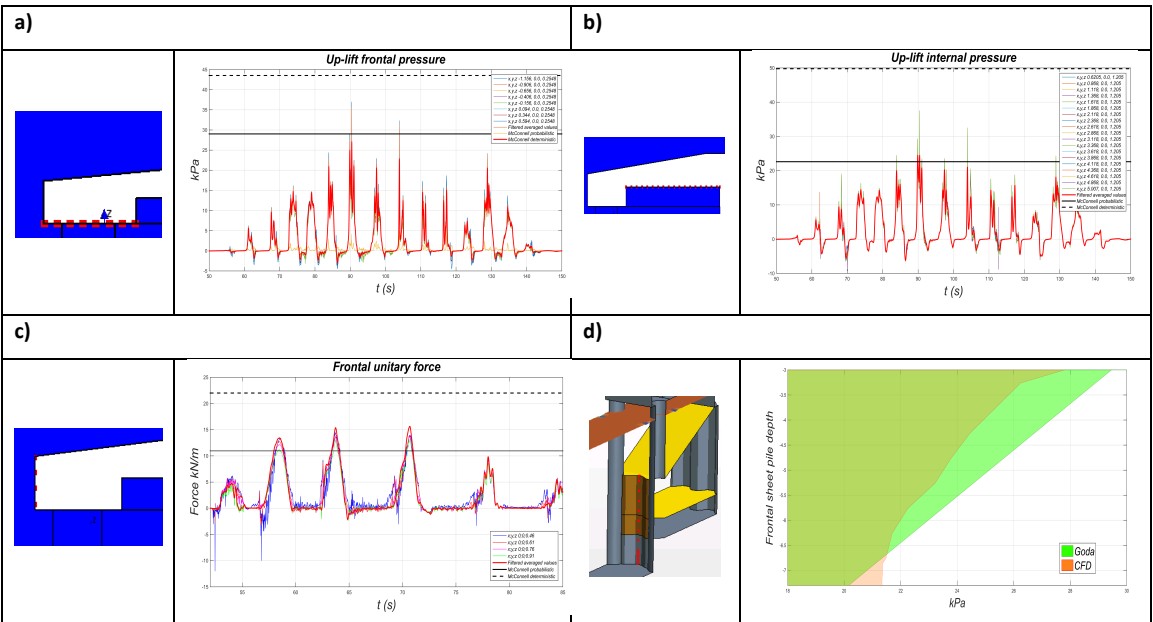

**Figure 20 Pressure gauges locations and measured time series**