# Peer review of "Wave simulation for the design of an innovative quay-wall: the case of Vlora's harbour"

_Natural Hazards and Earth System Sciences, 2016_

## Referee Comment (RC1) · Anonymous Referee #1 · 28 Jun 2016

I think the paper deserve to be published since it relies on a very applicable topic. The most interesting feature of this work is that it shows a clear and well-explained example of the usage of CDF, at the service of practical applications (in this case for performing wave climate). Furthermore, this work shows in simple way the potentialities given by the usage of a complex tool as CFD for maritime design studies, demonstrating to have the full domain of the matter. The paper is well written, the content clear and the figures properly described.

---

## Referee Comment (RC2) · Martinelli (Referee) · 20 Jul 2016

Review of manuscript "Wave simulation for the design of an innovative quay-wall: the case of Vlora's harbor" by Alessandro Antonini, Renata Archetti, Alberto Lamberti

The manuscript describes the application of a commercial CFD code to optimize the geometry of an absorbing quay wall. The necessary wave data are transferred to the harbour entrance with application of a 2D spectral wave model. The paper shows an example of effective use of advanced modelling for design purposes. The proposed concepts are sound. Proper and up to date references are given.

The stated objective of the paper is twofold, "i) showing application of sea situation estimation, through the use of wave hindcasted data, in order to properly define the design wave conditions for a new harbour, ii) presenting new approach for investigating

innovative non-reflective quay-wall".

General comments

The English language requires a revision, due to the presence of a several errors. Just a few examples are given below (see specific comments).

The proposed quay wall is considered "innovative" (see for instance the title). The reason why the structure is innovative should therefore be explained. I believe it is related to the construction technology, and therefore I suggest to provide a plan view of the structure.

The final optimal solution is in my opinion not very effective: in my experience (based on physical model testing) a better performance should be obtained using the "typical" absorbing quay configuration, i.e. with the rubble mound crest level above the mean water level, in order to induce wave breaking. May be artificial units such as tetrapods should be used instead of natural stones. I guess that most readers will be surprised to see that in the final configuration, your designed rubble mound crest is 1 m below the water level. I strongly suggest that the typical configuration is investigated, or at least discussed, to avoid possible critiques to your work.

Several figures are too small and the legend cannot be read even at maximum zoom.

Specific comments

Abstract

Include in the abstract the objective of the paper

Say where is Vlora (=Albania)

Add comma at the end of this paragraph: "Due to the particular geography and topography of the Vlora's Bay"→ "Due to the particular geography and topography of the Vlora's Bay,"

Rather than "non-reflective quay wall" I suggest "absorbing quay wall"

Rather than "allows the identification" I suggest "allows for the identification" (I'm not sure, though).

Introduction:

The rule «the singular form of nouns should be used when they function as adjectives» should be followed where appropriate (three year old boy, and not three years old boy, and therefore I believe waves conditions should be wave conditions; coastal structures design practice should be coastal structure design practice. . . this type of error is very frequent in the manuscript).

a widely number of methodology -> methodologies; (widely number? It looks over-stated)

"new concept of quay wall" A figure is needed pointing out the peculiarities. It is not clear what is the innovation.

an anchors and piles foundation system. . . a) it is incorrect in English; b) Where is the anchor?

Aim is showing? The English looks incorrect to me. Please correct.

Modeling of the armor slope

Can you give some sort of evidence that the numerical model is able to reproduce the dissipations due to possible breakings on the rubble mound (the mound that is present in the cross section of the quay wall)? This ability seems necessary for the optimisation of the absorbing quay structure.

Tables and figures

Table 4 presents a Tr=100 for two directions. Can you make a reference to a complete statistical analysis?

Figure 11: Innovative quay wall section: add a front view to clarify the structure

Figure 20: legends are not visible

[Figure]

---

## Author Comment (AC1) · 13 Sep 2016

Dear Reviewers,

our rebuttal to your comments and new version of the paper are available in the supplement zip file.

Please also note the supplement to this comment:
http://www.nat-hazards-earth-syst-sci-discuss.net/nhess-2016-168/nhess-2016-168-AC1-supplement.zip

---

## Author Comment (AC2) · 13 Sep 2016

Dear Reviewer,

we tahnk you for your appreciation of the paper and your comments. You will fin in the supplement an improved version of the paper.